Identification of a truncated splice variant of IL-18 receptor alpha in the human and rat, with evidence of wider evolutionary conservation

Booker Chris S. chris.booker@otago.ac.nz
Grattan David R.
Centre for Neuroendocrinology, Department of Anatomy, University of Otago , Dunedin , New Zealand
Patnaik Santosh
Electronic publication date: 2014 Sep 11
Publication date: 2014
Volume: 2
Electronic Location ID: e560
Received 2014 May 31; Accepted 2014 Aug 15
Copyright: © 2014 Booker and Grattan
Copyright year: 2014
Copyright holder: Booker and Grattan
License: This is an open access article distributed under the terms of the Creative Commons Attribution License, which permits unrestricted use, distribution, reproduction and adaptation in any medium and for any purpose provided that it is properly attributed. For attribution, the original author(s), title, publication source (PeerJ) and either DOI or URL of the article must be cited.
License URL: https://creativecommons.org/licenses/by/4.0/

Keywords: Interleukin-18, Toll-like receptor, Splice variant, Comparative genomics

Funding: Health Research Council of New Zealand Funding for this project was received from the Health Research Council of New Zealand (http://www.hrc.govt.nz). The project numbers are 08/076D and 11/1076D. The funders had no role in study design, data collection and analysis, decision to publish, or preparation of the manuscript.

==============================
Interleukin-18 (IL-18) is a pro-inflammatory cytokine which stimulates activation of the nuclear factor kappa beta (NF-κB) pathway via interaction with the IL-18 receptor. The receptor itself is formed from a dimer of two subunits, with the ligand-binding IL-18Rα subunit being encoded by the IL18R1 gene. A splice variant of murine IL18r1, which has been previously described, is formed by transcription of an unspliced intron (forming a ‘type II’ IL18r1 transcript) and is predicted to encode a receptor with a truncated intracellular domain lacking the capacity to generate downstream signalling. In order to examine the relevance of this finding to human IL-18 function, we assessed the presence of a homologous transcript by reverse transcription-polymerase chain reaction (RT-PCR) in the human and rat as another common laboratory animal. We present evidence for type II IL18R1 transcripts in both species. While the mouse and rat transcripts are predicted to encode a truncated receptor with a novel 5 amino acid C-terminal domain, the human sequence is predicted to encode a truncated protein with a novel 22 amino acid sequence bearing resemblance to the ‘Box 1’ motif of the Toll/interleukin-1 receptor (TIR) domain, in a similar fashion to the inhibitory interleukin-1 receptor 2. Given that transcripts from these three species are all formed by inclusion of homologous unspliced intronic regions, an analysis of homologous introns across a wider array of 33 species with available IL18R1 gene records was performed, which suggests similar transcripts may encode truncated type II IL-18Rα subunits in other species. This splice variant may represent a conserved evolutionary mechanism for regulating IL-18 activity.

Introduction

Interleukin-18 (IL-18) is a pro-inflammatory cytokine which has been linked with varying degrees of evidence to diseases as diverse as cardiovascular disease (Jefferis et al., 2011), asthma (Ma et al., 2012), inflammatory bowel disease (Siegmund, 2010; Matsunaga et al., 2011; Rivas et al., 2011), acute kidney injury (Ho, Fard & Maisel, 2010), and type 1 (Smyth et al., 2008) and type 2 diabetes (Thorand et al., 2005; Hivert et al., 2009), either through associations with IL-18 itself or with its receptor or binding protein. The IL-18 receptor belongs to the interleukin-1/Toll-like receptor superfamily, and is encoded by two genes—IL18R1 and IL18RAP—which encode a ligand-binding subunit (commonly known as IL-18Rα) and accessory protein (IL-18Rβ) subunit, respectively (Dinarello, 2009). The ligand-binding subunit, IL-18Rα, is also capable of binding a related cytokine, IL-37b (also known as IL-1F7b), to exert anti-inflammatory effects (Boraschi et al., 2011).

A mouse cDNA library generated by the Mammalian Gene Collection Program (Strausberg et al., 2002) was later found by Alboni et al. (2009) to include a cDNA sequence corresponding to a splice variant of IL18r1, formed by inclusion of part of an unspliced intron in the mRNA sequence. Alboni et al. (2009) named this variant type II IL18r1, with the reference sequence inheriting the nomenclature type I IL18r1, and presented evidence for its expression in the mouse brain. The reading frame resulting from inclusion of an unspliced intron introduces a stop codon shortly into the unspliced intron sequence. This splice variant would therefore be expected to translate into a truncated receptor with intact extracellular and transmembrane domains, but also a short cytoplasmic domain lacking the toll/interleukin-1 receptor (TIR) domain characteristic of IL-1 receptor family members. Given evidence suggestive of a role for IL-18 in the pathogenesis of a range of diseases in humans, we experimentally investigated whether a similar splice variant could exist in humans, finding evidence for an equivalent human type II IL18R1 transcript. Similar results were obtained in the rat, another commonly used laboratory rodent. IL-18 and its receptor are evolutionarily conserved across a wide range of species, and we therefore assessed whether transcription of homologous intron sequences in other species would be expected to generate truncated IL-18Rα subunits by assessing the predicted coding sequences of putative type II IL18R1 transcripts across a range of species with available IL18R1 gene records. These analyses suggest that the generation of truncated IL-18Rα subunits through alternative splicing may form a widespread mechanism of regulating IL-18 activity.

Materials and Methods

Ethics statement

All experimental procedures were approved by institutional ethics committees. The Animal Ethics Committee of the University of Otago, Dunedin, New Zealand, approved experiments involving rats (project number 58/07). For experimental procedures involving human samples, the only subject was the corresponding author (CB) and the research describes experimental work conducted on samples collected from the author. No written consent was obtained. This research was approved by the Human Ethics Committee of the University of Otago, Dunedin, New Zealand (project number 11/153).

Bioinformatics searches for prior evidence of human type II IL18Rα

In order to investigate whether an mRNA sequence incorporating intron 8-9 of the human IL18R1 reference sequence (Ensembl accession number: ENST00000233957, NM_003855.2) had been previously observed, BLAST searches were performed for cDNA or expressed sequence tags showing similarity to a 362 nt portion of the expected nucleotide sequence, covering the first 60 nucleotides of intron 8-9 preceded by the coding sequences of the two upstream exons (GACTCCAGAAGGCAAATGGCATGCTTCAAAAGTATTGAGAATTGAAAATATTGGTGAAAGCAATCTAAATGTTTTATATAATTGCACTGTGGCCAGCACGGGAGGCACAGACACCAAAAGCTTCATCTTGGTGAGAAAAGCAGACATGGCTGATATCCCAGGCCACGTCTTCACAAGAGGAATGATCATAGCTGTTTTGATCTTGGTGGCAGTAGTGTGCCTAGTGACTGTGTGTGTCATTTATAGAGTTGACTTGGTTCTATTTTATAGACATTTAACGAGAAGAGATGAAACATTAACAGGTAACACATATAATGCTGGAATTTCTTACCTTATGTTCTCATTAAGAAATCAGATAAATA) using a blastn search against the nucleotide collection (nr/nt) or expressed sequence tags (EST) databases limited to H. sapiens. Identified sequences of interest were further evaluated with the ‘Mapviewer’ tool from NCBI, and performing BLAST alignments against the human IL18R1 reference sequence (NM_003855.2/ENST00000233957) and predicted intron insert (intron 8-9 from ENST00000233957) to assess whether these sequences could represent a putative human type II IL-18Rα transcript.

Samples and RNA extraction

Experimental verification of human and rat type II IL18R1 transcripts was performed by RT-PCR on human and rat cDNA samples. Human whole blood was obtained from one of the authors (CB), a 32-year-old male of European descent, by venipuncture using a Vacutainer® tube containing potassium Ethylenediaminetetraacetic acid (EDTA) (BD Biosciences, USA). RNA was extracted using Zymo Research whole-blood RNA MiniPrep™ tubes (catalogue number R1020; Zymo Research, USA) without prior lysis of red blood cells, according to manufacturer’s instructions. Rat lung samples were obtained from three adult male Sprague-Dawley rats for verifying expression of a rat type II IL18r1 transcript, with samples collected fresh after decapitation, placed on dry ice, and transferred to a −80°C freezer until processing. Rats were obtained from the Hercus-Taieri Resource Unit, Dunedin, New Zealand which maintains a colony of Sprague-Dawley rats (Strain Code: 400, Crl:SD) originally sourced from Charles River Laboratories (Wilminton, MA, USA) and were from two litters with a mean age of 97 days (101, 89 and 101 days, rats A, B, and C, respectively) and mean weight 459 g (472 g, 443 g, and 461 g, respectively). Rats were housed in group housing with other littermates on 1/4″ Bed-o’Cobs® bedding from The Anderson Lab Bedding (The Andersons, Inc., Maumee, Ohio) and fed Rat and Mouse Cubes from Specialty Feeds (Glen Forest, Western Australia). Rats had not been involved in previous experimental procedures. Lung samples were homogenized using a Qiagen TissueLyser II (catalogue number 85300; Qiagen, Germany) in QIAzol Lysis reagent (catalogue number 79306) and RNA extracted with Qiagen RNeasy Plus Universal mini spin columns (product number 73404) according to manufacturer’s instructions.

Reverse transcription

Reverse transcription was performed using Superscript™ III reverse transcriptase (product number 18080-044; Invitrogen) in 20 µl reactions containing: 50 ng of random hexamers, approximately 700 ng of total RNA for human blood sample or 3,000 ng for rat lung samples, 10 nmol of dNTPs, 0.1 µmol of DDT, 4 µl of 5 × first strand buffer, 200 units of Superscript™ III (substituted for DEPC (diethylpyrocarbonate)-treated H2O for RT- control samples), 40 units of RNaseOUT™(product number 10777-019; Invitrogen) and DEPC-treated H2O as needed to complete reaction volumes. Random hexamers, dNTPs, and total RNA were first incubated at 65 °C for five minutes for annealing of hexamers, and after cooling the remaining reagents were added and samples incubated at 25 °C for 5 min, followed by 50 °C for 60 min, and reactions terminated by incubation at 70 °C for 15 min. For rat samples, following reverse transcription, 2 units of RNase H (product number 18021-014; Invitrogen) was added and samples incubated for 20 min at 37 °C to remove any remaining complementary RNA.

Primer design

Primers were designed to amplify human and rat IL18R1/IL18r1 reference sequence transcripts as shown in Table 1. For amplifying putative type II IL-18Rα transcripts, reverse primers were designed against the predicted inserted intron sequences. For rat samples, the same reverse primer as used by Alboni et al. (2009) for identifying type II mouse IL-18Rα was used with three nucleotides modified to match the homologous rat intron sequence. For human samples, the reference IL18R1 sequence (NM_003855.2/ENST00000233957) was truncated to exon 8 and combined with the first 300 nt of intron 8-9 (given that the murine type II IL-18Rα sequence incorporates the first 362 nt from the homologous mouse intron) to provide a predicted mRNA transcript sequence, and primers designed using Primer-BLAST (http://www.ncbi.nlm.nih.gov/tools/primer-blast/index.cgi) ensuring a reverse primer was placed in the intron insert (Table 1). For human samples, the reference IL18R1 sequence was used as a positive control to ensure expression of the IL18R1 gene could be detected in blood samples. For rat samples, primers against Actb were included as controls.

Table 1 Primer details for amplification of putative type II IL-18r1 transcripts in rat and human samples.

Species	Transcript	Primer	Primer sequence	Primer binding	Expected product size	
				mRNA/ DNA sequence:	nt		
Human	IL18r1	F primer	ACGCCGAGTTTGAA
GATCAGGGGT	ENST00000233957/ NM_003855.2	545–568	687 bp	
		R primer	CCCTGGGCAAAATCT
CCACAGCA	ENST00000233957/ NM_003855.2	1,209–1,231		
	IL18r1 type II
splice variant	F primer	ACGCCGAGTTT
GAAGATCAGGGGT	ENST00000233957/NM_003855.2	545–568	874 bp	
		R primer	ATACAGTTCCTG
GGCCCGAGCA	NT_022171.15 a	7,680,768–7,680,789		
Rat	IL18r1 type II
splice variant	F primer	CCAACGAAGAAG
CCACAGACA	NM_001106905.2	1,269–1,289	463 bp	
		R primer	AGCACGGGACA
TGTGAGGAGA	AC_000077.1 b	40,496,516–40,496,536		
	Actb	F primer	TACAACCTTCTT
GCAGCTCCTCCG	NM_031144.2	28–51	649 bp	
		R primer	TGTAGCCACGC
TCGGTCAGG	NM_031144.2	657–676		
Notes.

a Alternative R primer genomic coordinates: GRCh37, chromosome 2: 103,006,939–103,006,960.

b Alternative R primer genomic coordinates: RGSC3.4, chromosome 9: 39,652,714–39,652,734.

PCR and gel electrophoresis

Polymerase chain reactions (PCR) for amplification of human transcripts were performed using Platinum® Taq high fidelity and AccuPrime™ Taq polymerases (catalogue number 12567-012; Invitrogen, henceforth Platinum and Accuprime Taq). Reactions for human samples involving Platinum Taq contained 10 nmol dNTPs, 0.1 µmol MgSO4, 5 µl of 10 × buffer, 1 U of polymerase, approximately 100 ng of cDNA (assuming a 100% conversion of total RNA to cDNA during reverse transcription), 10 pmol each of forward and reverse primers, and made up to a total volume of 50 µl using DEPC-treated H2O. For reactions involving Accuprime Taq, 5 µl of 10 × ‘buffer I’, 10 pmol each of forward and reverse primers, approx 100 ng of cDNA, and 2 U of Accuprime Taq were combined with DEPC-treated H2O up to a total 50 µl volume.

Polymerase chain reactions for rat transcripts were performed in 50 µl reactions using Platinum Taq polymerase SuperMix (catalogue number 12567-012; Invitrogen), incorporating 45 µl of SuperMix (containing polymerase and dNTPs), 1 µl of DEPC-treated H2O, 20 pmol each of forward and reverse primers, and approximately 300 ng of cDNA.

Reverse transcription and PCR reactions were perfomed in polypropylene PCR tubes (catalogue number PCR-02D-L-C; Axygen, USA) and carried out on either a Biometra TProfessional Basic thermocycler (order number 070-701; Biometra, Germany) or an MJ Research Minicycler™ (Model PTC-150HB; MJ Research Inc, Watertown, MA, USA). For amplifications of human IL-18r1 reference and type II transcripts, thermocyclers were set to 94 °C for 2 min for initial denaturing, followed by 35 cycles of 94 °C for 30 s as a denaturing step, 55 °C for 30 s as an annealing step, and 68 °C for 1 min for extension. Reactions were terminated with a final extension step of 68 °C for 3 min and cooled to 10 °C for 3 min. For amplification of cDNA from rat samples, thermocycler settings were: 94 °C for 2 min for initial denaturing, followed by 35 cycles of 94 °C for 30 s as a denaturing step, 55 °C for 30 s for primer annealing, and 72 °C for 1 min for polymerase extension, followed by a final extension step of 72 °C for 5 min.

Gel electrophoresis of RT-PCR products was performed on 3% agarose gels (catalogue number 15510-027; Invitrogen Ultrapure™ agarose) using Qiagen GelPilot DNA loading dye (catalogue number 239901; Qiagen, Germany) and 100 bp DNA ladder (SKU#15628-019; Invitrogen). Images were captured on a digital camera connected to a Biometra BioDoc Analyzer running BioDoc Analyzer 2.1 software.

PCR product purification and Sanger sequencing

PCR products were purified for sequencing using Zymo Research DNA Clean & Concentrator™ −5 spin columns (catalogue number D4013; Zymo Research, USA) and sequenced at a commercial sequencing service (Genetic Analysis Services; University of Otago, Dunedin, New Zealand), which performs Sanger sequencing using an ABI 3730xl DNA Analyser with BigDye® Terminator Version 3.1 Ready Reaction Cycle Sequencing Kits. Chromatograms of sequencing products were used ‘as is’, with no attempt to manually correct ambiguous reads or base calling errors, and are included in File S1 (Chromatograms S1–S8). Sanger sequencing products and predicted IL-18r1 type II transcript sequences were compared in Geneious Basic (v5.6.; http://www.geneious.com/) using the ‘Geneious Alignment’ algorithm with default settings (65% similarity cost matrix, Gap open penalty of 12, Gap extension penalty of 3, assessed by global alignment with free end gaps). Consensus sequences from forward and reverse primer reads have been submitted to Genbank with accession numbers KM264374 (Human type II IL18R1 amplified with Accuprime Taq, shown in Fig. 2), KM264375 (Human type II IL18R1 amplified with Platinum Taq, shown in Fig. 2), KM264376 (Rat type II IL18r1 sequenced from Rat B, shown in Fig. S1), KM264377 (Rat type II IL18r1 sequenced from Rat C, shown in Fig. S1).

Bioinformatics assessment of human TIR domain

To assess the similarity between the predicted C-terminal portion of human type II IL-18Rα and other members of the IL-1 receptor family, protein sequences of the predicted human type IL-18Rα, interleukin-18 receptor 1 precursor (NP_003846.1), interleukin-18 receptor accessory protein precursor (NP_003844.1), interleukin-1 receptor type 1 precursor (NP_000868.1), interleukin-1 receptor-like 2 precursor (NP_003845.2), single Ig IL-1-related receptor (NP_001128526.1), X-linked interleukin-1 receptor accessory protein-like 2 precursor (NP_059112.1), interleukin-1 receptor accessory protein-like 1 precursor (NP_055086.1), interleukin-1 receptor-like 1 isoform 1 precursor (NP_057316.3), interleukin-1 receptor type 2 precursor (NP_775465.1), and interleukin-1 receptor accessory protein isoform 1 precursor (NP_002173.1) were aligned using the MUSCLE (MUltiple Sequence Comparison by Log-Expectation) online tool from the European Bioinformatics Institute (Cambridgeshire, UK).

Comparison of human type I IL18R1 (reference) and type II sequences

Given similarities in the predicted amino acid sequence of human type II IL-18Rα and the human IL-18Rα reference protein sequence, the respective intron and exon nucleotides encoding these amino acids were assessed by EMBOSS (European Molecular Biology Open Software Suite) Water alignment (European Bioinformatics Institute, Cambridgeshire, UK).

Assessment of putative type II IL18R1 nucleotide and type II IL-18Rα amino acid sequences across multiple species

In order to assess whether transcription of homologous introns into the IL18R1 transcript in other species would also result in truncated receptors, we searched for species with known IL18R1 genes by assessing homology with H. sapiens IL-18Rα through the Ensembl database. We identified 45 proteins which were further assessed for ambiguous amino acid and nucleotide sequences or other characteristics which would limit their assessment, as detailed in Table S1. For human IL18R1 transcripts, the Ensembl records IL18R1-201/ENST00000233957/ENSP00000233957 were chosen as the comparator as these show 1:1 identity to the NCBI sequences NM_003855.2/NP_003846.1 respectively. Two identified sequences in the Zebra Finch showed homology to the cytoplasmic domains of IL-18Rα, but sequences lacked any extracellular domains and whether these form functional receptors is unknown; these were therefore excluded from further analysis. Two splice variants were identified in the Tasmanian Devil which arise through the differential use of exons and encode proteins which differ in the cytoplasmic region immediately following the transmembrane domain, similar to the difference between IL-18Rα full length proteins and proteins predicted to be encoded by type II IL18R1 transcripts in mice, rats and humans. Therefore, these may represent type I (reference sequence) and type II IL18R1 splice variants in the Tasmanian Devil and were excluded from further analysis. Similarly, two splice variants were identified in the Turkey which arise through the differential use of exons, however the differences are limited and localized to the extracellular domain of the receptor, therefore only one of these was included in the analysis since both transcripts utilize the same exon structure in the portion which encodes the transmembrane and cytoplasmic domains of the receptor. After exclusions, 34 transcripts and amino acid sequences from 33 species were assessed by multiple sequence alignment to identify introns homologous to those unspliced in type II IL18R1 sequences in the mouse, rat and human, and to predict hypothesized C-terminal domains of type II IL-18Rα proteins. Multiple sequence alignments were performed using the MUSCLE tool available from the European Bioinformatics Institute (Cambridgeshire, UK) (EBI) and subsequently passed through the MView tool available from the European Bioinformatics Institute (Cambridgeshire, UK) for ease of visualization. Sequences were assessed through percent identities to human sequences as a reference and outputs are ordered anthropocentrically, i.e., according to similarity to human sequences. For assessment of insertion of an Alu transposable element in primate sequences, the human sequence surrounding the point of Alu insertion was run through the RepeatMasker online tool (Smit, Hubley & Green, 2012) to identify the Alu subfamily inserted, and the AluY sequence from the RepeatMasker site (http://www.repeatmasker.org/AluSubfamilies/humanAluSubfamilies.html) used in multiple sequence alignment against desired primate sequences using MUSCLE alignment. An analysis of evolutionary constraint across 29 mammalian species (Lindblad-Toh et al., 2011) includes overlap with 23 of the 33 species analysed here. Regions of high conservation within the human IL18R1 gene intron 8-9, shown in Fig. 6, were identified from the 29 mammals track of the University of California Santa Cruz (USCS) Genome Browser (http://genomewiki.cse.ucsc.edu/index.php/29mammals).

Nomenclature

Alboni et al. (2009) label the murine IL18r1 splice variant incorporating an unspliced intron ‘type II’ IL18r1, with the reference sequence becoming type I IL18r1. This nomenclature is continued here. In order to provide clarity between the splice variant mRNA transcript, which has been experimentally verified in the mouse (Alboni et al., 2009) and in the rat and human in the current work, and the predicted protein sequences encoded by the detected transcripts, throughout the text mRNA transcripts are referred to as either type I or type II IL18R1/IL18r1 (the mRNA products resulting from differential transcription of the IL18R1/IL18r1 gene), with protein sequences as type I or type II IL-18Rα (the mature full-length interleukin-18 receptor alpha reference protein or predicted truncated receptor protein, respectively).

Results and Discussion

We examined the Homo sapiens IL18R1 gene sequence (ENST00000409599) to first identify whether inclusion of an unspliced intron could give rise to a similarly truncated receptor. Transcription of intron 8-9 of the human IL18R1 reference sequence (Ensembl ENST00000233957, Genbank: NM_003855.2; intron 8-9 equivalent to hg19 chromosome 2: 103,006,678–103,010,928) would be expected to translate to a protein with a novel 22 amino acid C-terminal followed by a stop codon (Fig. 1), and thus generate a type II IL-18Rα protein lacking much of the cytoplasmic domain, as predicted for murine type II IL-18Rα.

Figure 1 Schematic diagram of IL18R1 reference sequence and aligned expressed sequence tags.

IL18R1 reference sequence (ENST00000233957/NM_003855.2) intron-exon structure aligned with three identified expressed sequence tags containing portions of intron 8-9. Shown above in boxed region is the expected protein coding sequence of a putative human IL18R1 splice variant incorporating intron 8-9 of ENST00000233957. Underlined nucleotides indicate those from the preceding exon 8. Codons are indicated by alternate grey shading of nucleotides. Incorporation of intron 8-9 would be expected to translate into a protein with a novel 22 amino acid C-terminal followed by a stop codon. First nucleotide shown in boxed region is nt 1132 from ENST00000233957/NM_003855.2, first amino acid is residue 370 from ENSP00000233957.1/NP_003846.1.

Figure 2 Schematic diagram of IL18R1 reference sequence and aligned expressed sequence tags.

IL18R1 cDNA from human blood was amplified by PCR using two polymerases, Platinum and Accuprime Taq as indicated, using the primer pairs shown in Table 1 to amplify the IL18R1 reference transcript (‘IL18R1’, expected product size 687 bp) or predicted human type II IL18R1 splice variant (‘type II’, expected product size 847 bp). ‘RT+’ and ‘RT−’ indicate PCR template generated with the addition of reverse transcriptase or without, respectively. Left hand lane shows 100 bp DNA ladder; brighter band towards the centre of the gel is 600 bp.

In order to investigate whether an mRNA sequence incorporating intron 8-9 had been previously observed, we performed BLAST searches for expressed sequence tags or nucleotide sequences using a 362 nt portion of the expected nucleotide sequence, covering the first 60 nucleotides of intron 8-9 preceded by the coding sequences of the two upstream exons. This revealed three sequences of interest with overlap between exon and intron nucleotides (BG541512.1, BG540341.1, and BG542027.1; Fig. 1). Of these, BG540341.1, and BG542027.1 show continuous readthrough of exon-intron sequences and could possibly represent genomic sequences. Of note, BG542027.1 represents a sequence beginning in exon 8 of the human IL18r1 reference sequence and aligns to the first 357 nt of the subsequent intron 8-9, up to nt 436 of BG542027.1, which is a total of 697 nucleotides and is noted to have high quality read up to nt 430. This aligned intronic region is similar to the length of inserted intronic sequence reported for murine type II IL18R1, which incorporates the first 362 nt of the homologous murine intron. BG541512.1, a cDNA sequence from human lung tissue submitted by the Mammalian Genome Collection Project, represented the longest of the identified sequences and showed reasonable similarity to the expected sequence of human type II IL18R1, incorporating nt 475-1135 of ENST00000233957/NM_003855.2 (representing a continuous read incorporating a portion of exon 3 through exon 8) and approximately 140 bp of intron 8-9 (represented schematically in Fig. 1). These data are consistent with the possibility that an alternate transcript incorporating intron 8-9 is present in humans.

In order to experimentally verify whether a human type II IL18R1 transcript could be detected, RT-PCR amplification of cDNA from human blood was performed, utilizing a forward primer placed within exon 4 and a reverse primer in intron 8-9 of ENST00000233957, in order to exclude the possibility of amplifying genomic sequences. Gel electrophoresis of RT-PCR products revealed a band of the expected size for a putative type II IL18R1 transcript (Fig. 2). Chromatograms showed a sudden drop in read quality around a CAG deletion which has been previously reported in humans (Watanabe et al., 2002), consistent with a heterozygous indel (Bhangale et al., 2005) (see File S1). Despite the drop in read quality, nucleotide sequences in lower quality read portions aligned well with the predicted human type II IL18R1 transcript sequence, and non-overlapping regions of high quality read on both sides of the indel from sequencing with forward and reverse primers aligned with the predicted sequence (not shown), demonstrating the RT-PCR product was indeed the predicted human type II IL18R1 transcript and incorporates previously intronic 5′ nucleotides of intron 8-9. Assessment of identity with the predicted human type II IL18R1 transcript is shown in Table 2A.

Table 2 Analysis of Sanger sequencing products for RT-PCR of human (A) and rat (B) IL-18r1 type II transcripts.

Sequencing products were compared with predicted IL-18r1 type II sequences derived from reference human and rat genes using Geneious Basic (v5.6.; http://www.geneious.com/). For human sequences, two deviations from the reference sequence were noted: rs1035130, and an nt 950 CAG deletion as reported by Watanabe et al. (2002). For rat sequences, a presumed SNP was identified which was consistent across both samples and in both sequencing directions at high read quality, present at nucleotide 39,652,571 of chromosome 9 (G > A; rat genome Baylor 3.4/rn4 build). In both cases, predicted sequences were edited prior to alignment to account for these variations in the tables below. Raw chromatograms are included in File S1.

(A)	
		Identities	Positives	Gaps	
Polymerase	Sequencing primer	%	n	%	n	%	n	
Platinum Taq	Forward	86	753/875	94	826/875	3	32/875	
	Reverse	88	773/877	94	832/877	3	32/877	
Accuprime Taq	forward	79	699/878	89	788/878	4	36/878	
	Reverse	83	736/877	91	805/877	3	33/877	
(B)	
		Identities	Positives	Gaps	
Sample	Sequencing primer	%	n	%	n	%	n	
Rat B	Forward	91	427/466	92	429/466	6	31/466	
	Reverse	91	427/466	91	428/466	7	35/466	
Rat C	Forward	91	427/465	92	430/465	6	31/465	
	Reverse	90	423/467	91	426/467	7	37/467	

Similar experiments conducted using cDNA isolated from lung tissue of Sprague-Dawley rats showed the presence of a type II IL18r1 splice variant transcript in this species as well (Fig. S1 and Table 2B). Whereas the mouse (Alboni et al., 2009) and rat type II IL-18Rα splice variants are predicted to encode 5 amino acids and result in truncated receptors which lack the TIR domain characteristic of members of the interleukin-1/toll-like receptor superfamily, the predicted sequence of human type II IL-18Rα was noted to encode amino acids similar to ‘Box 1’ of the TIR domain (Dunne & O’Neill, 2003) (Fig. 3A). Given the similarity in protein coding sequences, nucleotide sequences from intron 8-9 (encoding the C-terminal of the predicted human type II IL-18Rα) were compared with those of exon 9 (which encodes the beginning of the TIR domain in the human type I IL18R1 transcript) by pairwise alignment (Fig. 3B). Intron 8-9 exhibits similarity to exon 9, with a number of conserved codons. This suggests these two DNA segments may have arisen through duplication of a previous primordial single region.

Figure 3 Comparison of human type I and type II IL18R1.

(A) C-terminal ends of human IL-18Rα and predicted type II IL-18Rα amino acid sequences. First amino acid, underlined, corresponds to residue 370 from ENSP00000233957.1/NP_003846.1. Shown in red is the amino acid at which the IL-18Rα and predicted type II IL-18Rα sequences diverge, which is encoded across an exon/exon and exon/intron boundary in the case of the reference sequence and predicted type II sequence, respectively. Between the two sequences, similar residues are shown with the region of IL-18Rα corresponding to ‘Box 1’ of the TIR domain (Dunne & O’Neill, 2003) indicated. (B) shows pairwise alignment by EMBOSS Water (European Bioinformatics Institute, Cambridgeshire, UK) of the nucleotides from exon 9 of the human IL18R1 reference sequence (ENST00000233957/NM_003855.2) with those of intron 8-9. Codons are indicated by alternating grey shading, with the initial G nucleotide shown in brackets being the final nucleotide from exon 8 (nt 1135 of ENST00000233957/NM_003855.2). Above and below the respective sequences are shown the corresponding amino acid sequences.

IL-18 belongs to the IL-1 family of cytokines, and is highly homologous to IL-1, sharing similarities in processing, receptor function and downstream signalling (Dinarello, 1998; Dinarello, 2006). In the case of IL-1, a truncated ligand-binding receptor, IL-1R2 (aka IL-1RII), forms a “decoy receptor” lacking the intracellular TIR domain and unable to induce downstream signalling (Colotta et al., 1994). Multiple sequence alignment of members of the human interleukin-1 receptor family (Fig. 4) shows that IL-1R2 encodes an amino acid sequence with a C-terminal showing similarities to the ‘Box 1’ segment of the TIR domain. Similarly, the human type II IL18R1 transcript would be expected to encode a protein with a C-terminal showing similarity to the ‘Box 1’ segment, terminating just prior to the ‘Box 2’ segment of the TIR domain, suggesting the predicted type II IL-18Rα protein forms an truncated receptor for IL-18, analogous to the IL-1R2 receptor for IL-1.

Figure 4 Multiple sequence alignment of human IL-1 receptor family members.

The predicted protein coding sequence of human type II IL-18Rα aligned with IL-1 receptor family members. Shown are the amino acid residues surrounding the beginning of the TIR domain, where the sequence of type II IL-18Rα diverges from the reference sequence. Boxed regions are the ‘Box 1’ and ‘Box 2’ motifs present in the TIR domain.

The truncated inhibitory receptor for IL-1, IL-1R2, is found across a wide range of species, and given a conserved role for IL-18 and its receptor in many species, a similar inhibitory receptor for IL-18 could also be conserved throughout evolution. The previous findings of Alboni et al. (2009) showed the existence of a murine type II IL-18Rα and our findings of homologous splice variants in human and rat species suggested that insertion of an unspliced intron during transcription of the IL18r1 gene could form a mechanism of generating an inhibitory receptor for IL-18 across a more widespread range of species. We therefore asked whether transcription of homologous intron sequences would be expected to encode truncated splice variants of IL-18Rα across species for which gene records for IL18R1 were available.

We identified IL18R1 gene sequences for 33 species (Table S1, including human, rat and mouse) from which 34 transcripts and proteins were aligned to identify homologous intron sequences to those experimentally verified to be transcribed in mouse, rat, and human type II IL18R1 splice variants. Alignment of the nucleotide sequences of the preceding exon (equivalent to exon 8 of human IL18R1) and corresponding protein sequences showed a conserved predicted reading frame across species (Fig. S2). Analysis of predicted C-terminal amino acid sequences which would be translated from identified intron sequences is shown by multiple sequence alignment in Fig. 5. While the predicted mouse and rat protein sequences consist of a short C-terminal tail of 5 amino acids, in many species predicted sequences are a similar length to that predicted in humans with similarities to the ‘Box 1’ motif of the TIR domain. As expected, based on similarities in intron nucleotide sequences, predicted primate amino acid sequences were similar to the human sequence with the notable exception of the predicted Orangutan sequence, where an adenine in place of a thymine results in an adenosine codon in the Orangutan sequence where other members of the primate family show a stop codon (Fig. 6A).

Figure 5 Multiple sequence alignment of putative type II IL-18Rα C-terminal ends across multiple species.

Sequences are colour-coded as per MView (European Bioinformatics Institute, Cambridgeshire, UK): the human sequence is coloured according to residue properties, as indicated under ‘Key’, with residues from other species identical to human coloured with the same schema. Orangutan, Armadillo and Tarsier sequences are truncated by the indicated number of amino acids. Percentages reflect percent identity to human sequence as a reference.

Figure 6 Alignment of homologous intron sequences from IL18R1 across a range of species.

(A) When the reading frame is continued into the intron sequences, stop codons (red shading) are encountered shortly into the intron sequence, including a conserved stop codon position which resides within a region of high conservation (grey shading) as identified by Lindblad-Toh et al. (2011). Further downstream (B) an apparent conserved polyadenylation sequence (red shading, PolyA) is apparent, within a region of high conservation identified by Lindblad-Toh et al. (2011) (grey shading). A proposed transcription termination site (TTS) is indicated by the dotted line.

Of note, an Alu insertion is present in the homologous introns of many primate species assessed including the homologous human, orangutan, gibbon, gorilla, chimpanzee and macaque introns, but not in other species. Its presence in this group of primates, but not other primates assessed (Marmoset, Mouse Lemur, Bushbaby or Tarsier), would place its insertion at a last common ancestor in the Catarrhini parvorder following the separation of the Catarrhini and Platyrrhini parvorders, which is estimated to have occurred some 35 million years ago (mya) (Schrago & Russo, 2003) to 42 mya (Steiper & Young, 2006), at some stage in the mid- to late-Eocene. Assessment of the human intron sequence with RepeatMasker (Smit, Hubley & Green, 2012) identified the insertion as belonging to the AluY subfamily with 14 transitions, 2 transversions, and one gap of 2 nucleotides. Evident surrounding the complete AluY sequence is a target site duplication (TSD) and oligo(dA)-rich tail from Alu insertion. Upstream from the Alu insertion is a polyadenylation signal (PolyA; AATAAA) which is largely conserved across primates and also across the wider spectrum of species assessed (Fig. 6B). The expressed sequence tag BG542027.1 shows alignment of 357 nucleotides into the human intron 8-9 of ENST00000233957, at a point just downstream of this polyadenylation signal, which also aligns with the final nucleotide in the mouse type II IL-18r1 transcript (BC023240) before the polyadenine tail and thus is likely to represent the transcription termination site of human type II IL18R1. A recent analysis by Lindblad-Toh et al. (2011) identified regions of evolutionary constraint across 29 mammalian species and revealed five highly-conserved segments present in human IL18R1 intron 8-9 (excluding the initial 5′ highly conserved intronic nucleotides), four of which map to within or immediately adjacent to the human IL18R1 type II transcript. These represent two regions of overlap with identified stop codon sequences (one of which is depicted in Fig. 6A), another surrounding the polyadenylation signal (Fig. 6B), and a fourth downstream of the predicted transcription stop site, which we hypothesize is likely to play a role in transcription termination. This cross-mammalian conservation is likely to represent selection pressure acting at these loci.

We have found experimental evidence of the existence of a type II IL18R1 transcript in both human and rat species. In both of these cases, as well as the previously reported type II IL18r1 transcript in the mouse (BC023240) (Alboni et al., 2009), the splice variants arise from the insertion of unspliced homologous introns. Our analysis of the coding sequences of homologous introns in a wider array of species suggests that this mechanism of generating truncated versions of the IL-18Rα subunit could be evolutionarily conserved. Whether or not truncated splice variants are actually transcribed and translated in these species will need to be experimentally verified in each case. However, the regions of conservation identified here and previously by Lindblad-Toh et al. (2011) demonstrate the existence of selection pressure which is highly suggestive of a wider utilization of similar alternative transcripts beyond the mouse, rat and human.

While predicted rat and mouse type II IL-18Rα amino acid sequences consist of a short C-terminal tail immediately following the transmembrane domain, the predicted amino acid sequences in humans and many other species exhibit similarity to the initial region of the TIR domain incorporating the Box 1 motif. In the human (Fig. 4) it can be seen that IL-1R2 and the type II IL-18Rα splice variant appear to form homologous truncated receptors, with both arising in the ligand-binding subunit of the respective receptor complexes (as opposed to the accessory protein subunits), and both encoding short C-terminal domains with similarity to the ‘Box 1’ sequence of the TIR domain. Whereas the truncated IL-1R2 receptor is transcribed from a different gene than that encoding IL-1R1, in the case of IL-18Rα, a similar truncated receptor is predicted to be encoded by the use of a splice variant of the gene encoding the full-length receptor. Previously, IL-18 binding protein (IL-18 BP) has been noted as showing similarity to IL-1R2 in structure and function (Novick et al., 1999; Dinarello & Fantuzzi, 2003), with one study suggesting that they may share evolutionary origins (Watanabe et al., 2005). However our analysis suggests that, at least in terms of present day predicted protein sequences, IL-1R2 is in fact more similar to the predicted type II IL-18Rα. The similarity of the coding sequences of exon 9 and intron 8-9 in the human IL18R1 gene sequence (ENST00000233957/NM_003855.2) suggests that these gene sequences may have arisen through innovation, amplification and divergence (Bergthorsson, Andersson & Roth, 2007) of what was once a single genetic locus. Given similarities between intron sequences across the range of species assessed here, this explanation seems more likely than convergent evolution, and would also place such a duplication event at a very ancient timepoint in the evolutionary scale.

Exactly what role the Box 1-like motif plays in the relative functions of human versus rodent type II IL-18Rα is unknown. In the related IL-1R1, mutations in Box 1 have been shown to disrupt pro-inflammatory signalling (Slack et al., 2000), showing that this conserved domain is critical for normal receptor function. Crystal structures of the related Toll-like receptors 1 and 2 (TLR1/TLR2) and interleukin-1 receptor accessory protein-like 1 (IL1RAPL1) show that the initial Box 1 portion of these receptors forms a β-strand with the following residues forming an α-helix prior the second β-strand formed by the Box 2 motif (Xu et al., 2000; Khan et al., 2004). Particularly noteworthy are the findings of Khan et al. (2004) who assessed the crystal structure of IL1RAPL1, another member of the IL-1 family of receptors. In their analysis, constructs of human IL1RAPL1 lacking almost the entire cytoplasmic domain (including only amino acids 1–410) transfected into HEK293 cells were still able to activate JNK signalling. These constructs would include almost all of the Box 1 domain, terminating with the residues DAYLS (compare Fig. 4), and thus form receptors truncated at a similar point as IL-1R2 and the putative human type II IL-18Rα. While there is no evidence that IL-1R2 can induce JNK signalling, a number of studies have shown IL-18 is able to activate JNK signalling (Chandrasekar et al., 2005; Sahar et al., 2005; Seenu Reddy et al., 2010; Amin et al., 2010). The question which then arises is whether a truncated receptor for IL-18 would still be capable of inducing JNK signalling, as appears to be the case for a truncated IL1RAPL1 receptor. The work of Chandrasekar et al. (2005) showed in rat aortic smooth muscle cells that IL-18-induced JNK activation occurs downstream of a variety of intermediate signalling events including Myd88 activation, and likewise Adachi et al. (1998) showed a lack of IL-18-induced JNK activation in Myd88-/- cells; both provided support for IL-18 activation of JNK occurring downstream of Myd88 recruitment. Whether a truncated IL-18Rα subunit would be capable of forming dimers with IL-18Rβ and further complexes with Myd88 is unclear, but seems unlikely given mechanisms of action involving dimerisation of TIR domains.

Of particular relevance to these results are recent works investigating the processes controlling differential gene splicing. Given that multiple polyadenylation signals are often present within a gene and transcription occurs in a 5′ to 3′ direction, the question of how longer transcripts are generated when there are coexistent polyadenylation signals at more 5′ sites has been a topic of investigation. Studies have recently shown that U1 small nuclear RNA (snRNA), which is known for its role in splicing through recognition of 5′ splice sites, also plays a key role in suppressing the transcription of shorter transcripts associated with more 5′ polyadenylation signals, and thus regulates transcript length for genes with multiple polyadenylation signals (Kaida et al., 2010; Merkhofer & Johnson, 2012; Berg et al., 2012). Interestingly, autoantibodies to U1 snRNA occur in a number of autoimmune disorders (Breda et al., 2010; Kattah, Kattah & Utz, 2010), particularly mixed connective tissue disease and systemic lupus erythematosus, diseases in which altered IL-18 activity or its downstream effector, interferon-gamma (IFN-γ), have been observed (Bakri Hassan et al., 1998; Bodolay et al., 2002; Favilli et al., 2009). Berg et al. (2012) showed that depletion of U1 snRNA through RNA knockdown results in shorter transcripts corresponding to more 5′ polyadenylation signals. Given these findings, autoantibodies to U1-snRNA would be expected to result in an increase in the proportion of type II to type I IL18R1 transcription and a reduction in IL-18 activity (assuming type II IL18R1 transcripts form truncated receptors incapable of downstream signalling). In addition, rapidly dividing cells such as those of the immune system exhibit reduced U1-snRNA to transcript ratios (Merkhofer & Johnson, 2012) and shorter transcript lengths. Newly-dividing immune cells may therefore also express higher amounts of type II IL-18r1 transcripts, which could give rise to important differences in IL-18 responsiveness in dividing vs. mature cells. Neoplastic cells are also reknowned for their rate of cell division, and by the same mechanism a reduction in responsiveness to IL-18 through increased type II IL18R1 transcription may be of relevance for efforts to treat cancer with IL-18 therapies (GlaxoSmithKline, 2008; Srivastava, Salim & Robertson, 2010).

Previous research has identified a seemingly paradoxical increase in inflammatory signalling in cells lacking the IL-18Rα chain (Lewis & Dinarello, 2006; Nold-Petry et al., 2009). The anti-inflammatory cytokine IL-37 is able to bind the IL-18Rα chain, and the lack of signalling from IL-37 may underlie the reported phenotype of IL-18Rα-deficient cells. However, the existence of an inhibitory splice variant of IL-18Rα would also raise the possibility that this phenotype may derive in part from a lack of type II IL-18Rα. Whether IL-37 may in fact preferentially bind or act through a truncated IL-18Rα subunit is another possibility worth investigating.

The existence of an Alu insert in various primate IL18R1 gene sequences is in many ways unsurprising as it is well established that Alu sequences have undergone rapid expansion in the primate family (Batzer & Deininger, 2002; Cordaux & Batzer, 2009; Hwu et al., 1986). The presence of the Alu insert may be of relevance to the regulation of type II IL18R1 transcription in primates; Alu inserts have been shown to contain various regulatory regions, such as retinoic acid receptor motifs (Laperriere et al., 2007; Vansant & Reynolds, 1995) (which are largely conserved in the human IL18R1 gene sequence), and various studies suggest a role for Alu insertions in the regulation of gene expression in immune cells (Feschotte, 2008; Hambor et al., 1993) with Alu repeats hypothesized as giving rise to evolutionary changes in primates and humans (Cordaux & Batzer, 2009). While a number of studies report a role for Alu insertions in promoter sequences as regulators of gene function (Jacobsen et al., 2009; Pandey et al., 2011; Wang et al., 2011; Ebihara et al., 2002; Le Goff et al., 2003), there is evidence that insertion into other non-promoter regions is also capable of regulating gene transcription. For example, an intronic Alu element in the human CD8α gene regulates its transcription in T-cells (Hambor et al., 1993), Alu elements in the 3′ UTR of genes are involved in a mechanism of Staufen 1 (STAU1)-mediated mRNA decay (Gong & Maquat, 2011) and associate with lower levels of transcription (Faulkner et al., 2009), and an Alu repeat in the 3′ region of the human growth hormone influences its transcription rate (Trujillo, Sakagashira & Eberhardt, 2006). Of particular interest to the scenario in the human IL18R1 gene, where the Alu insert occurs downstream of the 3′ UTR, is a similar situation in the human APOA2 gene; in that case an Alu insert 305 nucleotides downstream of the polyadenylation signal of the APOA2 gene (Knott et al., 1985) (compared to 79 nucleotides from polyadenylation signal to 5′ TSD in human type II IL18R1) contains an SNP (rs12143180) leading to a MspI restriction polymorphism which is associated with lipoprotein levels (Civeira et al., 1992). These findings provide an example of an Alu element downstream of a gene regulating phenotype, and raise the possibility that the Alu element downstream of the human type II IL18R1 transcript could influence its transcription. Alu elements can also be subject to methylation (Xiang et al., 2010; Byun et al., 2012), and differential methylation could conceivably form a mechanism by which type II IL-18r1 transcription could be limited to select cells or stages of cellular development. If this Alu element does play a role in the regulation of type II IL18R1 transcription in humans, the lack of an equivalent Alu element in commonly used rodent laboratory animals suggests that this may form an important point of difference between IL-18 function in humans and rodents, and limit the generalization of results between these species.

One limitation of our study is that we have not verified whether human and rat type II IL18R1/IL18r1 transcripts are indeed translated into type II IL-18Rα proteins. This has also not been shown yet for murine type II IL18r1 transcripts. Antibodies for the detection of IL-18Rα will need to be carefully validated for their respective binding abilities of type I and the putative type II IL-18Rα proteins in order to enable experimentation into their presence in different tissue types. In addition, from our results the source of the identified type II IL18R1 transcripts has not been assessed in terms of examining specific cell types expressing this mRNA transcript. The type I reference IL18R1 transcript is expressed in a variety of cell types, and identifying which of these also express type II IL18R1 will be an important step in determining the physiological role of type II IL18R1. Neither have we specifically identified the transcription start or termination sites for human or rat type II IL18R1/IL18r1 transcripts. However, based on the cDNA record BG542027.1, alignment with mouse type II IL18r1, and the site of a putative polyadenylation signal, we conclude that the transcription termination site for human type II IL18R1 occurs 357 bp into the inserted intron sequence (see Fig. 6).

Conclusions

In conclusion, we have identified alternative transcripts of the human and rat IL18R1/IL18r1 genes, analogous to the previously reported type II IL18r1 transcript in the mouse (Alboni et al., 2009). These transcripts are likely to produce truncated proteins lacking most of the intracellular domain, which would be expected to result in altered signalling properties and thus may influence IL-18 activity in vivo. We also provided evidence that transcription of homologous intron regions in other species could give rise to similar truncated transcripts and that these genetic regions shown signs of selection pressure, indicating that this may be a mechanism of regulating IL-18 signalling which is conserved across different branches of the evolutionary tree. Given the apparent similarity between predicted protein sequences for these truncated IL18Rα isoforms and IL1R2, and hence a similar predicted function as a receptor which fails to elicit intracellular signalling, we suggest adopting the nomenclature IL18Rα2 to refer to these splice variant isoforms.

Supplemental Information

File S1 Raw chromatograms from Sanger sequencing of PCR products

Dataset S1—Chromatograms from Sanger sequencing of IL-18r1 type II PCR products shown in Figs. 2 and S1.

Click here for additional data file.

Figure S1 RT-PCR results for detection of a type II IL18r1 transcript in rat lung tissue

A 100 bp ladder is shown in the left lane, followed by reverse transcriptase positive (RT+) and negative (RT−) samples from 3 different rats (A)–(C). Expected product sizes for rat type II IL18r1 and Actb transcripts were 463 bp and 649 bp, respectively.

Click here for additional data file.

Figure S2 Alignment of nucleotide IL18R1 (A) and protein IL-18Rα (B) sequences from multiple species

Alignments show conserved reading frames across species, the site of insertion of an unspliced intron in type II IL18R1 nucleotide sequences (arrow and dotted line in (A)), and the corresponding point in protein sequences (arrow and dotted lines in (B)) at which the more C-terminal amino acids shown would be replaced by those encoded by the unspliced intron (shown in Fig. 5).

Click here for additional data file.

Table S1 Species with H. Sapiens IL18r1 homologues in Ensembl

Note that the two identified Guinea Pig sequences differ in their 3rd Ig domain (site of IL-18 binding) and are referred to by shorthand identifiers in the text and figures: * shorthand identifier Guinea Pig 22527 (for Gene ID: ENSCPOG00000022527), † shorthand identifier Guinea Pig 14114 (Gene ID: ENSCPOG00000014114).

Click here for additional data file.

Additional Information and Declarations

Competing Interests

Author Contributions

Human Ethics

Animal Ethics

DNA Deposition

The authors declare there are no competing interests.

Chris S. Booker conceived and designed the experiments, performed the experiments, analyzed the data, prepared figures and/or tables, reviewed drafts of the paper.

David R. Grattan conceived and designed the experiments, contributed reagents/materials/analysis tools, reviewed drafts of the paper.

The following information was supplied relating to ethical approvals (i.e., approving body and any reference numbers):

Human Ethics Committee of the University of Otago, Dunedin, New Zealand: Project number 11/153.

The following information was supplied relating to ethical approvals (i.e., approving body and any reference numbers):

Animal Ethics Committee of the University of Otago, Dunedin, New Zealand: Project number 58/07.

The following information was supplied regarding the deposition of DNA sequences:

Genbank: KM264374; KM264375; KM264376; M264377. Chromatograms from cDNA Sanger sequencing are supplied in Supplemental Information.

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
