# Peer review of "Identification of a truncated splice variant of IL-18 receptor alpha in the human and rat, with evidence of wider evolutionary conservation"

_PeerJ, doi:10.7717/peerj.560_

## Round 0.1 · original submission · Minor Revisions

· Academic Editor

Minor Revisions

The two reviewers of the manuscript have suggested very minor changes to the manuscript. Regarding the suggestion of reviewer 1, indeed it will be of help to readers if the new human and rat variant sequences can be provided. Please consider depositing the sequences in a nucleic acid sequence database such as NCBI GenBank, and providing the accession numbers in the manuscript. As for the suggestion of reviewer 2, I recommend using HGNC-approved gene symbols (http://www.genenames.org/) and providing aliases at the first mentions of the gene names. Thus, 'SIGIRR (IL1R8)' is okay, but 'IL1RII' can be replaced with 'IL1R2.'

·

Basic reporting

No comment. The paper complies will all that is requested.

Experimental design

This is an original work, clearly structured and conducted, based on a well-defined scientific question. Methods are exhaustively described. No other comments.

Validity of the findings

Reported data are solid and based on sound methodologies. Conclusions are fully in line with the data reported without any sort of overinterpretation.

Additional comments

The study “Identification of a truncated splice variant of IL-18 receptor alpha in the human and rat, with evidence of wider evolutionary conservation” by Booker and Grattan is a very interesting evaluation of the presence of transcripts, in man and rat, coding for an alternatively spliced truncated form of IL-1R5 (the binding chain of the IL-18 receptor complex). The lack of a complete intracellular domain leads to hypothesize that this receptor might have decoy functions, similarly to IL-1R2. The difference between the human alternative receptor (and those predicted in several other animal species) and the receptor identified in mouse and rat is the length of the intracellular domain, very short in rodents (suggesting a purely decoy function) but longer in man (the first TIR box is preserved) suggesting the possibility of an alternative functional role.
The authors speculate on the possible functions of these receptors across species, but correctly point out that no evidence is available regarding the cell types expressing the alternative receptor, or showing that the receptor is actually expressed as a protein.
The paper is very detailed and overall very interesting, in particular for the evaluation of receptor evolution and its biological significance. I only have a minor point. Since the authors have sequenced the human and rat alternative receptor cDNA fragments it would be of help to the reader to see the final consensus sequences, rather than a table listing the parameters and results of the sequence analysis, or in addition to it.

Reviewer 2 ·

Basic reporting

The figures are well conceived and very helpful in understanding of the novel IL-18R alpha truncation in humans are other species.

Experimental design

The data are generated using state of the art methods. The data are reproducible.

Validity of the findings

The data allow one to make the conclusions as stated by the authors. The comparable finding in thr human IL-1R type 2 is very consistent.
There is, however, a new nomenclature and IL-1RII is now written as IL-1R2. SIGIRR is now IL-1R8. IL-18R alpha should is correct. But for the truncation, thius Reviewer recommends IL-18Ralpha2

Additional comments

Very nice work

---

## Round 0.2 · accepted · Accept

· Academic Editor

Accept

The minor concerns raised by the reviewers have been adequately addressed in the resubmitted manuscript.